# Epidemiological description of 529 families referred for French transcultural psychotherapy: A decade of experience

Jonathan Lachal[1,2,3]*, Amalini Simon[1,2,3], Christine Hassler[3], Caroline Barry[3], Hawa Camara[1,2,3], Nelly Massari[1], Roberta Franchitti[1,4], Sann-Fou Mao[1,5], Tony Roy Edward[1,6], Laura Carballeira Carrera[1], Jeanne-Flore Rouchon[3,7], Marie Rose Moro[1,2,3]

1 AP-HP, Cochin Hospital, Maison de Solenn, Paris, France, 2 Université de Paris, PCPP, Boulogne-Billancourt, France, 3 Université Paris-Saclay, UVSQ, Inserm, CESP, Team DevPsy, Villejuif, France, 4 Sapienza Università di Roma, Rome, Italy, 5 Département Etudes Psychanalytiques, UFR IHSS, Université de Paris, Paris, France, 6 Institut de Psychologie, Université de Paris, Boulogne-Billancourt, France, 7 Assistance Publique–Hôpitaux de Paris (AP–HP), Hôpital Avicenne, Service de Psychopathologie, Université de Paris, France

* jonathan.lachal@gmail.com

**Data Availability Statement:** All relevant data are within the manuscript and its Supporting Information files.

## Abstract

### Background

Transcultural psychotherapy (TPT) is an original therapeutic method developed in various forms in France and several other countries in Europe as well as North America to address issues of migrant mental health care when psychosocial, economic, or cultural barriers hinder its accessibility and effectiveness. This study aims to describe the patients referred for TPT in Paris and its suburbs over the past decade, to examine intercultural differences and associations with social, demographic, and clinical variables, and to assess TPT in terms of patient adherence, attendance, and duration of care.

### Method

Retrospective study of 529 patients referred for TPT care, classified in three categories–no treatment, initiated treatment, engaged and continuing treatment. Collection and analysis of social, demographic, cultural, and clinical data, as well as of country of origin, duration of treatment, number of sessions attended (adherence), and number of sessions scheduled.

### Results

In all, 301 patients from 45 countries participated in an 11-month course of care lasting an average of 8 sessions. Most were children, accompanied by their families. The main psychiatric symptoms at the beginning of treatment were depressive, and the main cultural problem identified was the existence of a traditional theory explaining the illness in the family's culture. Patients kept 80% of their appointments for sessions, and attendance was not associated with socio-cultural or clinical variables.

**Funding:** The authors received no specific funding for this work.

**Competing interests:** The authors have declared that no competing interests exist.

## Conclusion

The high level of treatment adherence and attendance over time suggest that TPT is an effective method for addressing complex symptoms experienced by migrant families. Results highlighted the potential richness and originality of studies based on retrospective medical data.

## Introduction

Migrants and ethnic minorities are among the populations most highly exposed to health inequalities, especially in mental health [1]. Migrants face multiple psychosocial, economic, and cultural barriers that impede their access to health services [2–7]. While some authors characterize the Western mental health care system's response to the needs of ethnically diverse populations as an overall failure [8], others recommend culture-sensitive therapy programs and services aiming for the optimization of mental health care for immigrants [9]. Diagnostic issues are also more prominent among migrants, as cultural variations modify the symptoms and clinical presentation of the entire range of mental health problems, including common mental disorders such as depression or anxiety [10]. Migrants express distress in many different ways [11], show a greater tendency to somatization [12], present higher risks of chronic pain [13], and are at higher risk of misdiagnosis than nonmigrants [14].

Lastly, migrant patients prematurely terminate a large proportion of their mental health care treatments; likewise, ethnic minority status is one of the main risk factors for stopping treatment [15, 16]. The patient's ethnicity and the country have complex influences on this reality [15, 16]. Potential explanations include perceived racism, a preference for informal therapies outside the medical system, religious coping, and traditional explanations of illness and symptoms that do not match the explanations of host country therapists [16]. For example, prescriptions of antidepressants appear to vary according to patients' ethnicity [17]. Clinician factors contribute to these treatment differences, among them clinician bias, stereotyping, and uncertainty in clinical decision making and communication. Relevant patient factors include past adverse experiences with health professionals, opinions about health and its maintenance that differ from those of Western biomedical models, beliefs in external locus of control and other fatalistic notions, and the stigma surrounding mental disorders [17].

Transcultural psychotherapy (TPT) is an original therapeutic method developed in France and several European and North American countries to address these specific issues [18–23]. Its theoretical and methodological foundations rely on the work of Georges Devereux in ethnopsychiatry [24]. Devereux proposed a methodological principle—complementarism: the therapist must analyze the same material successively from different scientific perspectives, keeping them separate during the analysis and integrate these disciplines in the final interpretation. Devereux used psychoanalysis and anthropology, but the various scientific disciplines used today include history, linguistics, and sociology.

TPT was first successfully implemented in the Paris suburbs at the beginning of the 1980s and then spread outside Paris to diverse parts of France, French-speaking countries (e.g., Luxembourg, Belgium, Switzerland, French Canada), and other countries (such as Italy, Spain, Portugal, and Brazil). Patients encounter a group of therapists of multiple cultural and linguistic backgrounds. The size of the group is highly flexible and may vary from 2 to 10 or 15 therapists, depending on the patient's place of birth and life history. One of the therapists is called the main therapist. He/she leads the encounter and gives each speaker the floor. He/she may ask his/her co-therapists to provide comments, representations, symbolizations, metaphors, or

interpretations of any aspects of the patient's discourse. The group has four fundamental functions [18]: i) holding and psychic containment (general level); ii) recalling and emulating how traditional societies listen to and treat illness (cultural level); iii) enabling the process of decentering, as a materialization of otherness, and iv) proposing multiple different–and sometimes conflicting–ways of thinking about the illness. The systematic use of interpreters into and out of the family's mother-tongue completes the setting.

Transcultural care is a second-line intervention: all the patients referred have already undergone standard psychological treatment (first-line therapy) of variable durations. The referring professionals, most often psychiatrists or psychologists, refer patients for TPT when they find it difficult to build a trusting relationship and good communication, when they feel unable to deal with the cultural specificities of the patient's or family's representation of the illness, or when they doubt the correctness of their diagnosis or the patient's adherence to treatment and services, or both. There are five primary indications for TPT today. The first is the existence of a traditional theory that is impeding treatment. By a culturally-based theory, we mean a theoretical explanatory model of suffering and care different from the Western medical model, such as the paradigm of a hex on the family affecting one or more family members. The second is family conflict about cultural issues–in particular, questions about intergenerational conflicts and the *metissage* of two cultures—the ethnic-heritage culture and the new culture of the host society. In third place are cultural misunderstandings around illness and care that cause conflict between health professionals and the family. For example, the concept of chronicity in psychiatric disorders may not be understood by patients from a culture with circular rather than linear chronologies. A traumatic migration is the fourth indication. The fifth is the utilization of traditional care by the family, i.e., to combat witchcraft, when the medical team is worried about its effect on the robustness of the therapeutic alliance. A sixth, rarer, indication arises when a judge asks the group for an expert assessment of the transcultural aspect of an illness; in such cases, the patient is typically seen two or three times.

The therapeutic work comprises a dialogue between cultural meanings of the illness, traditional etiologies of suffering, and Western approaches to its care. The two therapeutic targets are the traumatic aspects of the migration and psychic cleavage.

The group meets the patients with their families for one-hour sessions every seven weeks. The core family is invited with any members of the extended family or other people to whom they feel close, as are the referring professionals. Patients come from diverse cultural backgrounds representative of the usual composition of immigration to France– 9.1% of the French population in 2014, with 31.5% from the European Union (EU), 30% from North Africa, 15% from sub-Saharan Africa, 15% from Asia, and 4% from European countries not members of the European Union [25].

TPT services have been available in Paris and its suburbs since 2008, provided by 2 teams at Avicenne AP-HP hospital (Bobigny) and 3 at Maison de Solenn, Cochin AP-HP hospital (Paris). Although several qualitative studies [26–29] have demonstrated the value of TPT, data about the patients referred for this treatment has not been published. Moreover, the significant clinical experience accumulated over the past decade strongly suggests that TPT may enhance the therapeutic alliance with migrant patients, reduce treatment dropout rates, and improve long-term prognosis. Because TPT was initially designed for patients from sub-Saharan Africa, on the model of the *talking tree* or *palaver tree* and a collective community approach to addressing issues [19], these positive outcomes may vary by cultural area of origin. We have found no study describing attendance and duration of TPT care.

To fill this gap, this study aims to: i) describe the patients referred for TPT at these two hospitals over the past 10 years; ii) examine intercultural differences and associations with socio-demographic, and clinical variables; and iii) assess the advantages of TPT by describing the

multiplicity of situations seen in consultation, analyzing data on adherence, attendance, and duration of the TPT care, and examine their associations with socio-demographic, cultural, and clinical variables.

## Materials and methods

This retrospective study describes a decade of TPT care at two hospitals.

### Participants

Pediatric and adult patients referred to two main outpatient departments in Paris (Maison de Solenn, Hôpital Cochin, APHP, and Service de psychopathologie de l'enfant et de l'adolescent, Hôpital Bobigny, APHP) for TPT between January 1, 2008, and May 31, 2018, comprised the study population. Data were collected between April and September 2018.

Participants were stratified into three groups according to the number of TPT sessions held with the patient and family (**Fig 1**):

- *Engaged treatment* group: three or more TPT sessions before ending

- *Initiated treatment* group: one or two TPT sessions before ending

- Referral rejected by the TPT referral manager: no TPT treatment, with the reason for rejection collected.

The decision of the three-session cut-off was made by a group of clinicians specialized in TPT on the basis of clinical arguments: the two first TPT sessions are devoted to the therapist group getting to know the patient and family, and vice versa, and to developing a therapeutic alliance that will encourage adherence and engagement and will enable effective treatment. From the third session on, these experts consider that therapeutic work has been engaged, is underway.

### Data collection

Data were collected from each patient's medical file, specifically, the referral request documents and the consultation reports. We collected:

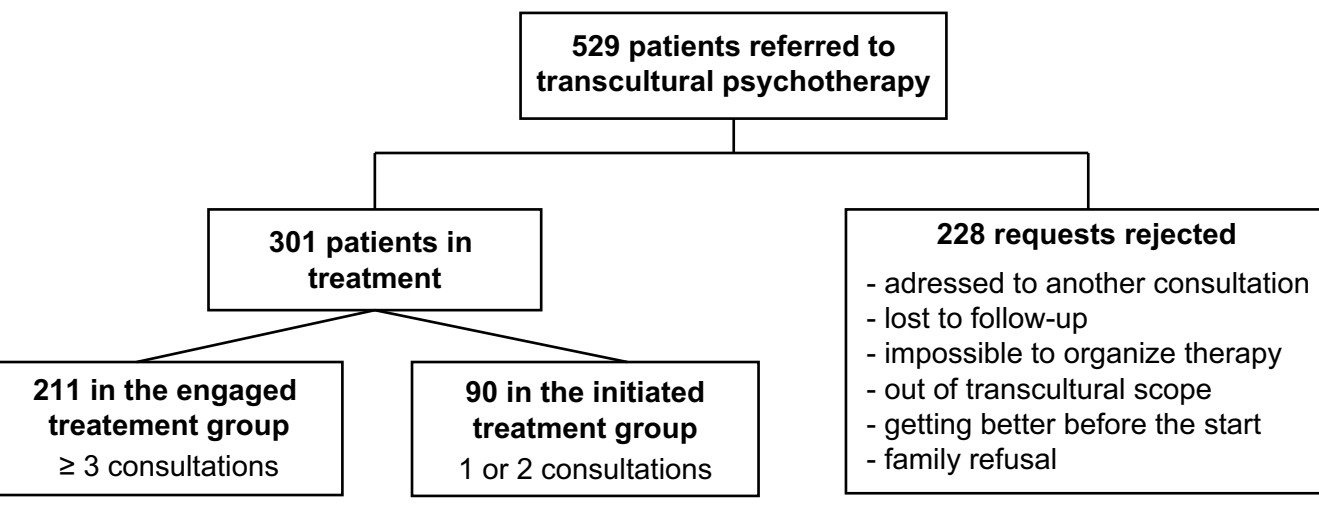

**Fig 1. Study flowchart.**

- Socio-demographic data: age, family composition

- Data on cultural origins: country of origin, languages spoken at home. Countries of origin were pooled into the following larger groups of *cultural areas*, both to guarantee anonymity and facilitate statistical analyses: sub-Saharan Africa; Asia; the Middle East and North Africa (MENA); the Caribbean; Europe–European Union non-members–; and Central and South America (only present in the rejected referral group). The sub-Saharan Africa population was chosen as the reference group because it was the largest and because TPT was initially designed to care for this population.

- Clinical data: psychiatric symptoms, psychological follow-up, cultural problem that led to referral for TPT, and TPT organization, including the presence of an interpreter and of any professionals from the referring first-line treatment organization. The psychiatric symptoms described here are those explicitly reported by the first line professional in the referral request.

Cultural problems were pooled into five categories for the principal current indications for TPT: i) *a cultural traditional theory of understanding illness*, *symptoms or care*; ii) *family conflicts about cultural issues;* iii) *cultural misunderstandings around illness and care* responsible for conflicts between health professionals and the family; iv) *a traumatic migration*; and v) *family use of traditional care.*

Administrative data were also collected from the hospitals' administrative software: date of first and last consultation, number of sessions attended, and total number of scheduled sessions.

Reasons for rejection of referrals were collected from the TPT referral request database.

We analyzed three *follow-up variables*: *TPT attendance* is defined by the percentage of sessions attended among those scheduled; the *follow-up duration* is the total period of follow-up; and the *number of sessions* is the total number of scheduled sessions, whether or not the patient/family attended.

This study was conducted according to the principles expressed in the MR-004 and approved by the National Commission for Informatics and Personal Liberties ('Commission Nationale de l'Informatique et des Libertés', CNIL).

## Statistical analysis

All data were entered in R software (3.6.0) for the statistical analysis.

First, univariate analyses were performed to describe the social, cultural, and clinical characteristics of the patients, and to assess the *follow-up variables*.

Next, we tested the associations between each *follow-up variable* (*TPT attendance*, *follow-up duration*, and *number of visits)* with the different socio-demographic, cultural, and clinical variables. Chi-square tests (or Fisher's exact tests when appropriate) and Student's t tests (or Wilcoxon tests if necessary) were performed for bivariate analysis. All statistical tests were bilateral and a *p*-value $< 0.05$ was defined as significant.

Finally, logistic regressions were performed to assess the associations between the clinical and socio-cultural variables and the cultural areas of origin.

## Results

The hospitals received 529 TPT referral requests from January 2008 through May 2018 (Fig 1).

## Description of the rejected referral group

In all, 228 referrals did not lead to TPT. Reasons for rejection were: a referral to another more appropriate type of treatment by the TPT referral supervisor (individual therapy or a more

**Table 1. Characteristics of the patients in the rejected referral group.**

| | Population % (N) |
|---|---|
| **Reasons for rejection** | |
| Referred for another treatment | 42.5 (97) |
| Lost to follow-up | 33.8 (77) |
| Impossibility to organize the therapy due to an acute situation | 12.7 (29) |
| *- no previous first-line therapy; referred for it* | *- 10.5 (24)* |
| *- medical acute disease* | *- 0.4 (1)* |
| *- acute family situation* | *- 1.8 (4)* |
| Barrier of language rather than culture | 8.3 (19) |
| Patient improvement before TPT start | 1.8 (4) |
| Family refusal | 0.9 (2) |
| **Cultural area of origin** | |
| Sub-Saharan Africa | 48.7 (91) |
| Asia | 19.8 (37) |
| MENA | 16 (30) |
| Caribbean | 7.5 (14) |
| Europe | 6.4 (12) |
| Central and South America | 1.6 (3) |
| **Total population** | 100 (228) |

%: percentage; N: population.

specific type of transcultural therapy, such as trauma therapy or for unaccompanied minors), loss to follow-up after submission of the first referral request (that is, patient was lost to follow-up in the period between the initial referral and its evaluation by the TPT referral supervisor), an acute situation impeding organization of the TPT, a problem requiring an interpreter but apparently not otherwise transcultural, patient improvement before the therapy starts, or opposition by the family. Table 1 describes the characteristics, including cultural area of origin, of the patients not treated by TPT.

## Participants' socio-demographic characteristics

Thus, 301 patients began TPT: 112 at Avicenne Hospital and 189 at Cochin Hospital. Among them, 211 had three or more consultations and were therefore classified in the engaged treatment group. Treatment ended for 90 patients before a third session was scheduled; they were included in the initiated treatment group (Table 2). Among these 301 patients, 29.5% were adults and 70.5% children, and among the latter, 6.6% (n = 14/213) were unaccompanied minors who came alone or with a social worker. Adults could come alone or with any members of their family. The mean (standard deviation, SD) number of children per family, among child and adult patients, was 3.13 (1.91).

The 301 patients who began treatment came from 45 different countries, the majority from sub-Saharan Africa, followed by Asia, MENA, Europe, and finally the Caribbean. They spoke 49 different languages besides French. The most frequent languages were Tamil (9.6%), Soninke (7%), Arabic (6.6%), Lingala (6.3%), and Bambara (5.3%).

Women were referred more often than men (four women for every man) and the mean age of adults was 36.3 years (SD = 12), with no significant difference between men (41.2, SD = 12.8) and women (35.1, SD = 11.6). First-generation migrants–the person who travelled– accounted for 93% of the adults.

**Table 2. Characteristics of the patients in treatment and comparison between those with initiated treatment and engaged treatment group (chi-square *p*-values).**

| | Total follow-up | Initiated follow-up group | Engaged follow-up group | *p* value |
|---|---|---|---|---|
| **Total Population (N)** | **301** | **90** | **211** | - |
| **Age** mean [SD, range] | | | | - |
| Children | **13.5 [5.2, 19]** | 13.1 [5.2] | 13.6 [5.2, 19] | |
| *Including UM* | ***16.7 [1.5, 5]*** | *15.5 [2.1]* | *16.9 [1.4, 4]* | |
| Adults | **36.3 [12, 49]** | 36.1 [10.8] | 36.4 [12.5, 49] | |
| **Age category** % (n) | | | | NS |
| Children | 70.8 (213) | 70 (63) | 71.1 (150) | |
| - *Including UM only* | - 6.6 (14) | - 3.2 (2) | - 8 (12) | |
| Adults | 29.2 (88) | 30 (27) | 28.9 (61) | |
| **Children's gender** % (n) | | | | NS |
| Boys | 52 (111) | 57.1 (36) | 50 (75) | |
| Girls | 48 (102) | 42.9 (27) | 50 (75) | |
| **Adults' gender** % (n) | | | | NS |
| Male | 19.3 (17) | 22.2 (6) | 18 (11) | |
| Female | 80.7 (71) | 77.8 (21) | 82 (50) | |
| **Cultural area of origin** % (n) | | | | NS |
| Sub-Saharan Africa | 57.5 (173) | 60 (54) | 56.4 (119) | |
| Asia | 19.9 (60) | 15.6 (14) | 21.8 (46) | |
| MENA | 13 (39) | 16.7 (15) | 11.4 (24) | |
| Europe | 5 (15) | 4.4 (4) | 5.2 (11) | |
| Caribbean | 4.7 (14) | 3.3 (3) | 5.2 (11) | |
| **Cultural problem** % (n) | | | | * |
| Traditional theory | 34.9 (105) | 32.2 (29) | 36 (76) | |
| Family conflicts | 23.3 (70) | 15.6 (14) | 26.5 (56) | |
| Cultural misunderstandings | 18.3 (55) | 20 (18) | 17.5 (37) | |
| Traumatic migration | 16.9 (51) | 18.9 (17) | 16.1 (34) | |
| Traditional care | 6.6 (20) | 13.3 (12) | 3.8 (8) | |
| **Presence of an interpreter** % (n) | | | | NS |
| Yes | 53.5 (161) | 45.6 (41) | 56.9 (120) | |
| No | 46.5 (140) | 54.4 (49) | 43.1 (91) | |
| **Referrer's profession** % (n) | | | | ** |
| Psychiatrist | 46.8 (141) | 40 (36) | 49.8 (105) | |
| Psychologist | 18.6 (56) | 21.1 (19) | 17.5 (37) | |
| Social worker | 18.6 (56) | 11.1 (10) | 21.8 (46) | |
| Other healthcare professional (general practitioner or other medical specialist) | 9 (27) | 15.6 (14) | 6.2 (13) | |
| Other (occupational therapist, speech therapist, et al.) | 7 (21) | 12.2 (11) | 4.7 (10) | |
| **Main psychiatric symptom** % (n) (Total patients/*Adults*/*Children*) | | | | NS |
| Depressive symptoms | 37.5 (113)/*56.8 (50)*/29.6 (63) | 37.8 (34)/*59.3 (16)*/28.6 (18) | 37.4 (79)/*55.7 (34)*/45 (30) | |
| Psychotic symptoms | 21.9 (66)/*19.3 (17)*/23 (49) | 24.4 (22)/*25.9 (7)*/23.8 (15) | 20.9 (44)/*25.9 (16.4)*/22.7 (34) | |
| Externalized symptoms | 19.3 (58)/*5.7 (5)*/ 24.9 (53) | 14.4 (13)/*7.4 (2)*/17.5 (11) | 21.3 (45)/*4.9 (3)*/28 (42) | |
| Developmental symptoms | 11 (33)/*1.1 (1)*/15 (32) | 15.6 (14)/*0 (0)*/22.2 (14) | 9 (19)/*1.6 (1)*/12 (18) | |
| Anxious or traumatic symptoms | 10.3 (31)/*17 (15)*/7.5 (16) | 7.8 (7)/*7.4 (2)*/7.9 (5) | 11.4 (24)/*21.3 (13)*/7.3 (11) | |

*(Continued)*

**Table 2.** (Continued)

| | Total follow-up | Initiated follow-up group | Engaged follow-up group | *p* value |
|---|---|---|---|---|
| **Presence of the first line professional** % (n) | | | | * |
| Yes | 76.1 (229) | 67.8 (61) | 79.6 (168) | |
| No | 23.9 (72) | 32.2 (29) | 20.4 (43) | |

UM: unaccompanied minor; %: percentage; N: population; NS: non-significant

*: p<0.05

**: p<0.01

***: p<0.001.

Nearly half the children (48.5%) were the oldest child in their family, while second children accounted for 20.7% of the population younger than 18 years, third children 15.7%, and those ranked fourth to ninth in the family 15.1%. Mean age was 13.5 years old (SD = 6.9). Boys and girls were referred at an equal rate (1:1), but the boys referred were significantly younger (12.5, SD = 5.37) than the girls (14.5, SD = 4.8) (*p*<0.01). Second-generation migrants–born in France of parents born abroad–accounted for 75% of the children. Oldest siblings were born in France in 67.8% of the cases. Of pediatric patients born abroad, 81.8% were the last child of the family born abroad.

The 14 unaccompanied minors had a mean age of 16.7 years, and three fifths of them were boys. They came from Asia (50%), sub-Saharan Africa (42.9%), and the Caribbean (7.1%). We were unable to collect data on family composition in these situations.

Finally, 22 patients had mixed families, that is, with one parent born abroad, the other in France. The mother was the foreigner in 12 of these families, the father in 10. They came from sub-Saharan Africa (13), MENA (4), Asia (3), and Europe (2).

## First line psychological follow-up

TPT is a second-line intervention, and all of the patients had previously received mental care. Those in care when TPT began were being seen at that time by one (45%), two (35%) or three or more (16%) types of professionals, specifically by a psychiatrist (38%), psychologist (22%), social worker (22%), physician (11%), speech therapist (4%), or occupational therapist (3%). The median length (1st quartile-3rd quartile) of first-line follow-up before the start of the TPT was 18 months (9–36). Finally, only 4% of the patients had no first-line intervention in mental health when TPT began.

Most referrals for TPT came from psychiatrists, psychologists, or social workers (Table 2). The patients were then being seen at a hospital (38%), a medical/psychological center for out-patient mental health care (23%), a mother and child protection center (6%), or a school (4%).

## Cultural problems

TPT is indicated when an identified cultural issue impedes the effectiveness of usual care. The main cultural problem identified in our sample was the existence of a cultural traditional theory that explained the illness, symptoms or care, followed by family conflicts about cultural issues, and cultural misunderstandings around illness and care (Table 2 and S1 Table).

## Main psychiatric symptoms

The principal categories of psychiatric symptoms at the beginning of the follow-up were depressive, psychotic, or externalized (Table 2). They differed somewhat between adults and

children–adults experienced mainly depressive symptoms (56%), while children experienced quite similar proportions of externalized (25%), psychotic (23%), and depressive symptoms (29%).

## First consultation

Overall in both hospitals, an average of 28.9 (7.1) new patients began treatment each year.

First-line therapists are warmly invited to participate in at least the first sessions between the patient and the TPT team, and in 75% of cases, one such professional was present (Table 2): a psychiatrist (36%), social worker (30%), or psychologist (34%). In one third of these cases, 2–4 members of the first-line intervention team accompanied the families.

The assistance of an interpreter in the families' native language is always strongly suggested as an important aspect of treatment. Nevertheless, it is not always possible to find an interpreter for the necessary language. Furthermore, some families refuse to have an interpreter present. In all, an interpreter was present for slightly more than half of the first sessions. Interpreter presence rates were similar for children and adults and for men and women.

## Cultural area (Table 3)

We tested associations between cultural area of origin and socio-demographic and clinical variables.

We found no statistically significant associations between cultural area of origin and either age–i.e. adult vs children–, or gender among children or adults.

Multivariable regression models between cultural area of origin and cultural problems showed that Caribbean (OR = 7.8, CI = 2.1–51.4) origins were associated with a higher risk of

**Table 3. Logistic regression of child and adult gender and age, cultural problems, presence of an interpreter, and main psychiatric symptom on cultural area of the entire follow-up population (odds ratios and odds ratios and their 95% confidence intervals).**

| | Overall *p*-value | Reference = Sub-Saharan Africa (n = 173) | | | |
|---|---|---|---|---|---|
| | | Asia (n = 60) OR [CI] | MENA (n = 39) OR [CI] | Europe (n = 15) OR [CI] | Caribbean (n = 14) OR [CI] |
| **Generation** (Ref = children) | NS | 0.9 [0.8–1] | 1 [0.8–1.1] | 1.2 [0.9–1.5] | 1.2 [1–1.6] |
| **Children gender** (Ref = girls) | NS | 1 [0.5–1.9] | 1.2 [0.5–2.6] | 0.9 [0.2–4.1] | 1.3 [0.3–6.6] |
| **Adults gender** (Ref = women) | NS | 0.3 [0–2] | 0.9 [0.1–4] | 1.4 [0.2–7.3] | 0 |
| **Cultural problem** | | | | | |
| Traditional theory | *** | 0.2 [0.1–0.4]*** | 0.5 [0.2–1]* | 0.1 [0–0.5]* | 7.8 [2.1–51.4]** |
| Family conflicts | NS | 1.4 [0.7–2.8] | 1.4 [0.6–3] | 3.5 [1.1–10.3]* | 0.7 [0.1–2.6] |
| Cultural misunderstandings | *** | 3.2 [1.6–6.3]*** | 0.7 [0.2–1.9] | 3 [0.9–9.1] | 0 |
| Traumatic migration | NS | 1.4 [0.6–2.8] | 1.1 [0.4–2.6] | 0.8 [0.1–3] | 0 |
| Traditional care | * | 1 [0.2–3.3] | 4.7 [1.7–13] ** | 0 | 0 |
| **Presence of an interpreter** (Ref = no) | *** | 10.4 [4.7–26.2]*** | 1.8 [0.9–3.6] | 1.2 [0.4–3.5] | 1 [0.3–3.1] |
| **Main psychiatric symptom** | | | | | |
| Depressive symptoms | NS | 1.1 [0.6–2] | 1.2 [0.6–2.5] | 0.9 [0.3–2.6] | 1.3 [0.4–3.9] |
| Psychotic symptoms | NS | 0.5 [0.2–1.1] | 0.6 [0.2–1.4] | 0.2 [0–1.1] | 1.1 [0.3–3.6] |
| Externalized symptoms | NS | 0.9 [0.4–1.8] | 1.1 [0.4–2.5] | 2.8 [0.9–8.4] | 0.3 [0–1.7] |
| Developmental symptoms | NS | 2.2 [0.9–5] | 0.8 [0.2–2.6] | 2.5 [0.5–8.8] | 0 |
| Anxious or traumatic symptoms | NS | 1.3 [0.5–3.2] | 1.4 [0.5–4] | 0 | 2.7 [0.6–9.7] |

OR: Odds Ratios; CI: Confidence intervals; NS: non-significant

*: p<0.05

**: p<0.01

***: p<0.001; Ref = reference.

*traditional theory problems*, compared with patients from MENA (OR = 0.5, CI = 0.2–1), Asia (OR = 0.2 CI = 0.1–0.4), and Europe (OR = 0.1 CI = 0–0.5). In contrast, Asian or European origins were associated with higher odds ratios for *cultural misunderstandings* (respectively OR = 3.2, CI = 1.6–6.3, and OR = 3, CI = 0.9–9.1), whereas MENA origins were associated with a higher odds ratio of *traditional care problems* (OR = 4.7, CI = 1.7–13).

The distribution of psychiatric symptoms did not differ between cultural areas of origin, with the logistic regressions showing no statistically significant associations.

Finally, logistic regressions showed a statistically significant association between Asian origin and an interpreter's presence at the first session (OR = 10.4, CI = 4.7–26.2).

### Follow-up

Of all the sessions scheduled, 70% involved patients participating in treatment underway for 3 or more sessions. Engaged treatment was statistically significantly associated with the presence of first-line professionals at the first session, with each of the cultural problem variables, and with the profession of the person making the referral (Table 2).

The average number of total sessions was 8.3 (6.9) overall, and 10.6 (6.8) for the patients engaged in treatment (Table 4). Total sessions were not significantly associated with any of the variables considered.

Attendance was high, with patients attending an average of 80% (SD = 19.6) of their sessions; the rate was higher for those engaged in treatment, at 82.3% (SD = 16). Attendance was not significantly associated with any of the study variables.

Finally, the median duration of follow-up was 11 months (5–24) and rose to 18 months (10–29.5) for those engaged in treatment. Again, this outcome variable was not significantly associated with any of the other variables studied.

### Discussion

This study is the first to describe the clinical, cultural, and family characteristics of migrant patients seen in TPT in France.

Their countries of origin are broadly representative of the countries from which most immigrants come to France [25]: Africa, Asia (mainly former French colonies), and Europe. Nevertheless, the distribution is significantly different from that of French immigration overall (43.8% from Africa, 36% from Europe, 14.5% from Asia, and 5.6% from America & Oceania [25]). The low proportion of European patients is easily explained by the cultural proximity of other European countries to France, so that standard psychiatric care is generally effective. Moreover, as already said, TPT was initially designed for patients from sub-Saharan Africa [19], which may explain the overrepresentation of African families in TPT. The particularly high rate of Tamils may be explained by the recent migration of refugees from Sri Lanka, who settled in large numbers around one of the participating hospitals (Avicenne hospital) [30]. Finally, only three patients from the Americas were referred, and none was found to have a transcultural indication. This undoubtedly reflects the low rates of immigration from the Americas to France.

Children and adolescents account for most of the patients referred. The experience of migration–encompassing the travel as well as the entire process of settlement and accultura-tion–stresses them more strongly than their parents and is correlated with more difficult psy-chological adjustment for them [31], despite the potential benefits to children of living in two cultures (e.g. a bicultural identity). Several vulnerability factors have been uncovered and differ between first- and second-generation migrant children. First-generation migrant children are at high risk of exposure to violence in their country of origin (war, discrimination, poverty,

**Table 4. Characteristics of engaged treatment group–number of visits, mean attendance and treatment duration–according to gender, age, cultural area, cultural problems, presence of an interpreter, referrer's profession, psychiatric symptoms, and presence of the first-line professional.**

| | Number of visits mean [SD] | Attendance mean [SD] | Treatment duration (months) median [1stQ-3rdQ] |
|---|---|---|---|
| **Total Population** (NB: Engaged *follow-up group only*) | **10.6 [6.8]** | **82.3 [16]** | **18 [10–29.5]** |
| **Age** | | | |
| Children | 10.4 [6.6] | 81.7 [16.2] | 17 [9.5–28.5] |
| Adults | 11.1 [7.3] | 83.9 [15.5] | 21 [10–31.3] |
| **Children's gender** | | | |
| Boys | 10.1 [6.5] | 83 [16] | 16 [11–28.8] |
| Girls | 10.7 [6.8] | 80.3 [16.4] | 17 [9–28.5] |
| **Adults' gender** | | | |
| Male | 11.6 [7.1] | 89.3 [14.6] | 29 [18–36] |
| Female | 11 [7.4] | 82.7 [15.6] | 21 [9.5–29.5] |
| **Cultural area of origin** | | | |
| Sub-Saharan Africa | 10.5 [6.8] | 81.6 [16] | 19 [10–28.8] |
| Asia | 11.4 [7] | 84.7 [15] | 19 [11–34] |
| MENA | 10 [7] | 82.6 [15.1] | 13 [8–21.5] |
| Europe | 9.6 [7.2] | 84.6 [19.3] | 18 [6.8–23.8] |
| Caribbean | 11.3 [6.1] | 76.9 [19.2] | 21 [8–32] |
| **Cultural problem** | | | |
| Traditional theory | 10 [6.2] | 83.2 [15.8] | 19 [9–32] |
| Family conflicts | 11 [6.7] | 81.4 [15.3] | 19 [10–29] |
| Cultural misunderstandings | 9.1 [5.9] | 82.4 [17.6] | 13 [7–30.5] |
| Traumatic migration | 13.5 [8.9] | 81.1 [16.3] | 21 [12–28] |
| Traditional care | 8.6 [3.7] | 85.9 [16] | 14 [10–23] |
| **Presence of an interpreter** | | | |
| Yes | 11 [6.9] | 82.9 [15.9] | 19 [10.25–29] |
| No | 10.1 [6.7] | 81.5 [16.1] | 18 [9–31] |
| **Referrer's profession** | | | |
| Psychiatrist | 10.8 [6.7] | 82 [15.2] | 18 [11–28.5] |
| Psychologist | 10.8 [6] | 80.5 [14.8] | 23.5 [11–34] |
| Social worker | 9.8 [7.5] | 83.9 [18.1] | 13.5 [8–22] |
| Other healthcare professional | 12.7 [8.7] | 78 [18] | 20.5 [12.5–29.5] |
| Other | 8.5 [5.3] | 91 [14.6] | 12.5 [7.3–31.8] |
| **Main psychiatric symptom** | | | |
| Depressive symptoms | 11.1 [6.1] | 79.9 [16.6] | 21 [10.3–29] |
| Psychotic symptoms | 9.6 [6.1] | 84.3 [13.4] | 17 [10–29.5] |
| Externalized symptoms | 10.3 [7.7] | 82.7 [15.9] | 14 [7–22] |
| Developmental symptoms | 9.4 [5] | 87.3 [15.3] | 22 [9.8–33.3] |
| Anxious or traumatic symptoms | 12.3 [9.3] | 81.9 [18.5] | 18 [13.8–34.5] |
| **Presence of the first-line professional** | | | |
| Yes | 10.3 [6.4] | 82.3 [15.4] | 17 [10–27.5] |
| No | 11.9 [8.3] | 82.6 [18.1] | 23 [8.5–33.5] |

SD: standard deviation; 1stQ-3rdQ: first and third quartiles values.

and crime) or during migration, and they frequently face poverty, racism, and discrimination in the host country [32, 33]. Family support decreases psychological symptoms, but does not decrease the experience of depression [5]. Cumulative exposure to traumatic events is associated with a broad range of psychological problems in migrants, and family exposure to

violence is more strongly associated with psychological outcomes in children than the children's own exposure is [34]. Finally, first-generation migration children are especially exposed to acculturative stress, which can cause family problems or increase their parents' stress levels as well. The burden of acculturative stress on internalizing disorders such as depression and suicidal ideation is particularly high for these children [35–37]. Second-generation migrant (SGM) children, on the other hand, are at higher risk of intergenerational conflict. Once a family immigrates, its members live in two cultures: their "old country" ethnic culture and the new culture of the host society. Recent immigrant families typically continue to retain many of the cultural values of their original, or ethnic, culture. The second-generation children acquire their ethnic heritage through their parents and relatives and the culture of the host country from their peers [38, 39]. The cleavage between these two cultural worlds produces a specific psychological vulnerability. The transcultural paradigm supports the idea that the child often bears the suffering of the entire family [40]. Parenthood in migration requires the ability to show resilience over adversity to ensure a better future for one's children. In many cases, migrant parents are more concerned with subsistence than living, and they have neither the time nor the money to deal with their own psychological suffering. This is particularly true of men. Children, on the contrary, are brought or referred to psychological care by their schools or the child protection agency. These points may explain both the higher TPT referral rates for minors and the sex-ratio differential among adults. The latter result may also be explained by the more general underutilization of mental health services by men–native born or immigrant, in France and in many other countries.

The different generations of migrants represented by children and adults forecasts upcoming issues in mental health care for migrants: the number of immigrants arriving in Europe is stable or decreasing, but the number of their offspring is rising, and they now account for one third of the births in France [41]. Nonetheless, we found no association between generation of migration, sibling birth rank, or rank in relation to migration (the last child born in the country of origin or the first child born in France) and symptoms that result in a TPT referral. The child referred was the oldest in nearly half the cases. We might hypothesize that the oldest is the most exposed to acculturative stress and intrapsychic conflicts between the cultural expectations of parents and of peers. The success of the oldest in reconciling and mixing the culture may well make it easier for the younger children.

Although the presence of an interpreter is strongly recommended for all TPT sessions because of their important role in the therapeutic alliance [42–45], our results showed that an interpreter is present in only three-fifths of all sessions. The use of an interpreter was most frequent among patients coming from Asia. It is reasonable to speculate that more patients come from former French colonies in the other cultural areas (North Africa, French West Indies, and West and Central Africa) and thus speak better French; it might even be their native language. Furthermore, the Asian migration is the most recent, and migrants from there have spent the least amount of time in France and thus are likely those with the poorest mastery of the language.

The distribution of cultural problems may reflect the specificities of each cultural area: the proximity between French and other European cultures may explain the higher level of family conflicts and the lower level of traditional theories. Inversely, traditional theories around illness in sub-Saharan Africa and the Caribbean differ quite markedly from those in the West and rely most often on external causality–for example, mental suffering may be explained by possession or witchcraft. This point may explain the high proportion of issues involving traditional theories in these two subgroups. This result is in line with the literature: for example, Lanzara et al. found conflicting results about differences in prevalence of somatization between cultural groups [46].

A substantial proportion of referral requests (two-fifths) did not lead to TPT. In half of those cases, however, the families were redirected to more specific psychotherapeutic settings, for example, trauma care, or one of the specific TPT teams for unaccompanied minors or international adoption consultations, or individual transcultural treatment [20, 47, 48]. TPT is a relatively expensive group therapy, with lengthy interviews and many health professionals; it must be reserved for complex situations. Furthermore, those specific transcultural teams are most appropriate for their specific symptoms or situations. Because the referring partners are generally unfamiliar with their specificities, most referrals are addressed to the main TPT consultation.

Among the 301 patients who began TPT, 70% engaged ongoing care that lasted for three or more sessions. In our clinical experience, the initiated treatment of one or two sessions encompasses very diverse situations: dropouts, but also inappropriate referrals, and quite often, issues that could be resolved quickly. TPT is most often a brief therapy: a little work on reconciling Western and traditional approaches to suffering and care is often enough to resolve the main difficulties and refer patients back to their first-line treatment. Because our study is based on retrospective data, we were unable to collect any indicators of the effectiveness of these brief interventions. Nevertheless, it appears clear that the early dropout rate among the patients is lower than in any other psychotherapeutic setting, despite the additional risk factors among migrant patients. Dropout rates are high in standard psychotherapy, even in naturalistic studies [16]: rates may vary from 25 to 75% depending on the design and the definition of dropping out [49–51]. For patients with ethnic minority status, dropout rates may rise from 40% to more than 60% [16]. Moreover, many significant predictors of dropping out identified in robust studies are frequently encountered in migrant patients: lower socioeconomic status, a younger mother, a poorer therapeutic relationship, or lower perceived relevance of treatment (mainly as perceived by parents) [16, 52, 53].

Furthermore, attendance rates were very high in our clinical sample–attendance exceeded 82% at sessions among the patients engaged in established treatment. The small number of studies reporting attendance rates in various psychotherapy and other therapeutic settings makes direct comparisons quite difficult. Betancourt et al. found a similar retention rate of 82% in a randomized trial testing a family-based mental health promotion among Somali Bantu and Bhutanese refugees [54]. But the comparison is subject to caution because of the difference in the design of the study (interventional study vs observational study). Nevertheless, standard adherence to appointments was reported at around 60% [55]. Again, belonging to an ethnic minority, as well as disadvantaged socioeconomic factors, normally predict higher no-show rates. This was not the case in our results.

One way of understanding these results about involvement and attendance is to hypothesize that TPT reduces the risk of poor adherence by improving the family's opinion about the relevance of treatment [16]. The higher the risk of divergence between patients or parents and clinicians in understanding the illness and in choosing therapeutic options, the higher the dropout and no-show rates. In particular, it has been shown that treatment adherence and attendance are lower when the parent-clinician relationship is not good and more specifically when their conflict concerns the relevance of the treatment [16, 53, 56]. This could lead not only to higher no-show rates, but also to the therapist being "turned off" by the parent [53]. Ethnic minority patients and parents are more highly exposed to this risk, because cultural differences complexify common issues about understanding care [19, 31]. As a result, in general, in most psychotherapeutic settings, ethnic minority status is one of the main risk factors for dropping out of mental health treatment [16] and may lead to poorer therapeutic outcomes [53, 57, 58]. TPT, however, aims to create a dialogue between the clinicians' and the families' theories of the illness and to work on building compromises between the two cultural worlds–

the therapists' medical world and the family's traditional world. In this way, TPT may reduce barriers associated with participation in treatment and thus improve patients' attendance and adherence.

The families referred because of issues related to family investment in traditional care had lower rates of engaged treatment than the others. This result is consistent with the preceding paragraph, as families are likely to find it more difficult to invest a psychotherapeutic treatment when they are already involved in another promising care. The therapeutic work on finding a compromise between the two forms of treatment remains touchy, even in TPT.

Finally, we found a positive association between engaged treatment and some factors reinforcing dialogue and teamwork with families and first-line therapists. When the first-line coordinator, that is, psychiatrist or social worker, has made the referral for TPT or when the first-line team is present during the TPT sessions, the risk of early dropout is reduced, as already shown in qualitative investigations (Carretier E, Grau L, Mansouri M, Moro M, Lachal J, Qualitative assessment of transcultural psychotherapy by adolescents and their migrant families: Subjective experience and perceived effectiveness, 2019, unpublished).

## Strengths & limitations of the study

The main strength of the study is the originality and the size of the population: this is the first study to describe the migrant population with psychological disorders referred for TPT in France. The study covered two sites and included the entire population referred to TPT over a decade. The relatively large sample (529 patients and families) allowed us to perform statistical analyses across cultural areas of origin.

Among the limitations is the pooling of patients into six "cultural" areas of origin, which erases most of the specificity of our population and means that each cultural area encompasses many different cultural realities. We nonetheless think that this choice was the best possible: it does preserve some cultural specificity at the same time that it enables descriptive and statistical analysis. Using the country of origin would have created problems about anonymity and statistical measures. At the same time, in view of the cultural heterogeneity in many countries, it would not have been any more culturally meaningful. There are, for example, more than 200 different ethnic groups with their own language in Cameroon. At the same time, two ethnic groups with very different cultures may share a language.

No descriptive data were available for the rejected referral group, making comparison with initiated and engaged treatment groups impossible.

Because of the small size of two subsamples (Europe and Caribbean) and thus insufficient statistical power, interethnic differences probably could not be detected. Furthermore, the lack of data about rejected patients limits the possibilities of understanding the higher rates of children treated by TPT.

Finally, and most important, the data we could measure—attendance, duration of treatment, or number of sessions—say nothing about the content of the therapy or its effectiveness [59]. Hypotheses about an association between attendance, adherence, and success of the therapy need to be examined in relation to clinical practice. A prospective study might better explore these links.

## Conclusion

This is the first study describing the cultural and clinical characteristics of a large sample of 529 patients and their families referred for TPT in France. The results highlight the variety and richness of observed clinical situations. The overrepresentation of children may result from two factors: their greater psychological vulnerability, but also greater vigilance about the

suffering of children. The high level of patient attendance, adherence, and engagement in treatment suggest that TPT is an effective method for addressing the complex symptoms experienced by migrant families. This hypothesis will be clinically explored with an upcoming RCT protocol [60].

Our study also highlighted the richness and originality of the examination of retrospective medical data and should encourage the multiplication of such studies to increase awareness of the types of clinical care offered and the patients seen in psychotherapy.

## Supporting information

**S1 Table. Characteristics of the entire treatment population.**
(DOCX)

## Acknowledgments

We thank Madeleine GUILEE and Floriane CRINE, our invaluable medical archivists and secretary, Dr Carole GIACOBI & Dr Adrien LENJALLEY for their participation in the data collection, Jo Ann CAHN for the translation, and all the therapists and trainees from five transcultural consultations.

## Author Contributions

**Conceptualization:** Jonathan Lachal, Amalini Simon, Christine Hassler, Caroline Barry, Hawa Camara, Roberta Franchitti, Sann-Fou Mao, Tony Roy Edward, Laura Carballeira Carrera, Marie Rose Moro.

**Formal analysis:** Jonathan Lachal, Christine Hassler, Caroline Barry.

**Investigation:** Jonathan Lachal, Amalini Simon, Hawa Camara, Nelly Massari, Roberta Franchitti, Sann-Fou Mao, Tony Roy Edward, Laura Carballeira Carrera, Jeanne-Flore Rouchon.

**Methodology:** Jonathan Lachal, Amalini Simon, Christine Hassler, Caroline Barry, Hawa Camara, Marie Rose Moro.

**Project administration:** Hawa Camara, Jeanne-Flore Rouchon, Marie Rose Moro.

**Resources:** Nelly Massari.

**Software:** Jonathan Lachal, Christine Hassler.

**Supervision:** Jonathan Lachal, Christine Hassler, Caroline Barry, Jeanne-Flore Rouchon, Marie Rose Moro.

**Validation:** Jonathan Lachal, Amalini Simon, Jeanne-Flore Rouchon, Marie Rose Moro.

**Writing – original draft:** Jonathan Lachal, Amalini Simon, Christine Hassler, Caroline Barry, Hawa Camara, Nelly Massari, Roberta Franchitti, Sann-Fou Mao, Tony Roy Edward, Laura Carballeira Carrera, Jeanne-Flore Rouchon, Marie Rose Moro.

**Writing – review & editing:** Jonathan Lachal, Amalini Simon, Christine Hassler, Caroline Barry, Hawa Camara, Nelly Massari, Roberta Franchitti, Sann-Fou Mao, Tony Roy Edward, Laura Carballeira Carrera, Jeanne-Flore Rouchon, Marie Rose Moro.

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
