## [Decision Letter · Decision Letter 0]

13 Feb 2020

PONE-D-19-34239

An epidemiological description of 529 families treated by the French method of Transcultural Psychotherapy: A decade of experience

PLOS ONE

Dear Dr. Lachal,

Thank you for submitting your manuscript to PLOS ONE. After careful consideration, we feel that it has merit but does not fully meet PLOS ONE’s publication criteria as it currently stands. Therefore, we invite you to submit a revised version of the manuscript that addresses the points raised during the review process.

We would appreciate receiving your revised manuscript by March 13, 2020. To enhance the reproducibility of your results, we recommend that if applicable you deposit your laboratory protocols in protocols.io, where a protocol can be assigned its own identifier (DOI) such that it can be cited independently in the future. For instructions see: http://journals.plos.org/plosone/s/submission-guidelines#loc-laboratory-protocols

We look forward to receiving your revised manuscript.

Kind regards,

Stephan Doering, M.D.

Academic Editor

PLOS ONE

Journal Requirements:

Reviewers' comments:

Reviewer's Responses to Questions

**Comments to the Author**

1. Is the manuscript technically sound, and do the data support the conclusions?

Reviewer #1: Yes

Reviewer #2: No

Reviewer #3: Yes

2. Has the statistical analysis been performed appropriately and rigorously? 

Reviewer #1: Yes

Reviewer #2: I Don't Know

Reviewer #3: Yes

3. Have the authors made all data underlying the findings in their manuscript fully available?

Reviewer #1: Yes

Reviewer #2: Yes

Reviewer #3: Yes

4. Is the manuscript presented in an intelligible fashion and written in standard English?

Reviewer #1: Yes

Reviewer #2: Yes

Reviewer #3: Yes

5. Review Comments to the Author

Reviewer #1: The study is well-done, scientifically sound and addresses an important topic for which the authors should be congratulated. However, I strongly recommend authors to enrch the article with more scholarly references.

See my suggestions below:

INTRODUCTION

Authors should provide the readers with a broader background and state-of-art of the topic under study: for example, when they speak of mental and psychological issues affecting migrants, they could discuss the following articles Somatic perception, cultural differences and immigration: results from administration of the Modified Somatic Perception Questionnaire (MSPQ) to a sample of immigrants. Bragazzi NL, Puente GD, Natta WM. Psychol Res Behav Manag. 2014 Jun 12;7:161-6; Effects of acculturation, coping strategies, locus of control, and self-efficacy on chronic pain: study of Chinese immigrant women in Italy - insights from a thematic field analysis. Re TS, Bragazzi NL, Siri A, Cisneros Puebla C, Friese S, Simões M, Candau J, Khabbache H. J Pain Res. 2017 Jun 6;10:1383-1390; Front Psychol. 2019 Jan 17;9:2792. doi: 10.3389/fpsyg.2018.02792. eCollection 2018. A Clinical-Psychological Perspective on Somatization Among Immigrants: A Systematic Review. Lanzara R, Scipioni M, Conti C.

DISCUSSION

Authors could enrich the discussion comparing their results with the literature and quoting relevant articles, especially systematic reviews such as: Front Psychol. 2019 Jan 17;9:2792. doi: 10.3389/fpsyg.2018.02792. eCollection 2018. A Clinical-Psychological Perspective on Somatization Among Immigrants: A Systematic Review. Lanzara R, Scipioni M, Conti C.

Reviewer #2: I have reviewed the manuscript “An epidemiological description of 529 families treated by the French method of Transcultural Psychotherapy: A decade of experience”. The study aims to describe the patients referred for Transcultural Psychotherapy (TPT) in Paris and its suburbs over the past decade and to assess TPT in terms of patient adhesion, attendance, and duration of care. However, the overall purpose of this paper and its broader relevance remain unclear. This together with several other shortcomings further described below and most importantly the lack of correlation between results, discussion and conclusion, leads me to recommend rejection of this paper.

Introduction: The paper concerns TPT, a therapy that the authors state is widely used in French speaking countries. The references cited (ref. 16-19) are however all written by the last author of the present study and are all except one in French which limit the accessibility to a non-French speaking audience. In the present paper, there a no description of the content of TPT or reference to manuals. This makes it difficult to compare the result of the present study to others and limits the clinical usefulness.

Materials and Methods: The authors divide their population into three categories: “Referral rejected”, “initiated treatment” and “effective treatment”. However, the only thing that separates the two latter groups are the number of sessions attended. I find it highly problematic that the authors mix up treatment adherence and effect, a confusion that remains throughout the paper. Therefore, would strongly advice against using the word “effective”, as there are no measures of treatment effect in this paper.

Results: The result section is lengthy, and it is again difficult to clearly see the relevance of part of the provided information. For example, on p. 9, the author provides a very thorough description of the origin of different groups of children, only to conclude that cultural area of origin was not associated with age or gender.

Main psychiatric symptoms are mentioned on. p. 11, but it is not anywhere stated how information about these were obtained (for example whether if they origin from self-ratings or clinical interviews) which is of major importance for the reliability of these data.

Table 4 (p. 12) solely concerns the “effective” treatment group, again with no explanation of the rationale behind this presentation of findings.

Discussion: The third and fourth section of the discussion (P. 13-14) are comparing first and second generation of migrant children and has seemingly little to do with the results of the present study. From p. 15 and onwards the author starts discussion their own results – but the argumentation in often unclear. For example, on p. 15, bottom section the author states: “it appears clear that rate among the patients with effective treatment is lower than in any other psychotherapeutic setting”. However, the “effective” treatment group are defined by persons who have continued treatment past 2nd session and hence it makes no sense to say that the early dropout in this group was low.

The authors furthermore explain the higher rates of children in TPT with the sentence “the experience of migration stresses them more strongly than their parents” (p. 13), a statement that is not at all accounted for by neither the data (where 75% of the children were second generation migrants) or by the general literature. The overrepresentation of children in the present study would likely be due to other factors such as referral/rejection patterns (though little information is given on the rejected patients, it could be the case that the majority of those referred to individual therapy would be adults) and/or the content of the intervention.

As for attendance the authors state that it is hard to find comparable studies that provide attentions rates. However, there are studies to be found with comparable populations and attendance rates (Betancourt et al., 2019). The discussion of the reasons for the high attendance rate seems quite speculative and does not mention the possibility that attention rate is (relatively) high because the therapy is brief and not very intensive (meetings every 7th week!).

The limitation section is short, and the authors seem unaware of the major limitations of their study.

Conclusion: None of the conclusions drawn seem to be supported by the data. Again, it is stated that the overrepresentation of children is due to their greater psychological vulnerability and greater vigilance about the suffering of children, conclusions that cannot be drawn based on available data (please see above). Even more problematic is the link the authors draw between attendance and effect. The author states (p. 17): “The high level of attention, adherence, and engagement in treatment indicates that TPT is an effective method addressing the complex symptoms experienced by migrant families.” This conclusion is downright wrong since no measures of treatment effect are applied in the present study. While adherence may to some extent be correlated to treatment satisfaction, it has nothing to do with effect and patients can easily be satisfied and adherent with their treatment, despite having limited outcomes (Buhmann et al., 2018).

Based on the above, I find that the paper does not fulfill PLOS One publication criteria no. 4 (Conclusions are presented in an appropriate fashion and are supported by the data) and furthermore lacks the quality and relevance to a broader audience expected from a PLOS ONE publication, which is why I recommend rejection.

References:

Betancourt TS, Berent JM, Freeman J, Frounfelker RL, Brennan RT, Abdi S, Maalim A, Abdi A, Mishra T, Gautam B, Creswell JW, Beardslee WR (2019) Family-Based Mental Health Promotion for Somali Bantu and Bhutanese Refugees: Feasibility and Acceptability Trial. J Adolesc Heal. doi:10.1016/j.jadohealth.2019.08.023

Buhmann CB, Carlsson J, Mortensen EL (2018) Satisfaction of trauma-affected refugees treated with antidepressants and Cognitive Behavioural Therapy. Torture J 28: 118–129.

Reviewer #3: Comments to the Author

Thank you for the opportunity to review the manuscript entitled “An epidemiological description of 529 families treated by the French method of Transcultural Psychotherapy: A decade of experience”.

General Comments

The authors have carried out a retrospective study of 529 patients referred for the Transcultural Psychotherapy in Paris according to the French method. They classified the patients in three categories (no treatment, initiated treatment and effective treatment) and analyzed the differences between the initiated treatment and effective treatment group as well as the intercultural differences concerning clinical factors like reasons for attendance, main psychiatric symptoms and socio-demographic variables.

The study concerns an important issue in the field of mental health of migrants. The manuscript is well written, the methods and statistical analyses are appropriate. The results seem consistent and plausible. The discussion is clear and coherent, the conclusions are adequately drawn from the demonstrated data.

I have only some minor points that should be considered in the revision:

Title:

As some of the patients had been classified to the group “no treatment” because they rejected the TPT for different reasons, the following statement in the title is not correct: “529 families treated by the French method of Transcultural Psychotherapy” -> please modify.

Abstract:

- Please add further aims: examination of intercultural differences and associations with socio-demographic and clinical variables.

- Please delete the word “descriptive” in the first sentence of the method because it can be misunderstood due to the fact that the presented study is not only a descriptive study because it reports not only descriptive statistics.

Introduction:

- p. 2: The authors state in the third sentence that “some authors characterize the Western mental health care system´s response to the needs of ethnically diverse populations as an overall failure”. But there are also other authors who recommend culture-sensitive (therapy) programs and services aiming for the optimization of mental health care for immigrants, e.g. in Canada or in Germany or other Western countries. Please add this information to the introduction with an example for such a culture-sensitive approach in the Western mental health care system.

- p. 4: please report how large is the proportion of immigrants in the population in France and from which countries come the three largest immigrant collectives.

Results:

- Table 1, last line: please place 228 to the right – to the column below N.

- Table 2: please change N to n in the columns Initiated follow-up group and Effective follow-up group.

- Table 2: please add the information regarding mean age (SD and range) of the children and adults and analyze gender-specific differences.

- Table 2: please provide an example for the categories Other healthcare professional and Other.

- Table 2: please add the data concerning the main psychiatric symptoms for children and for adults.

- p. 9: “the sex ratio was 25”; … “around 100” – it is not clear what you mean, please reformulate.

- p. 9: “the mean age 36.3 years (12).” – what do you mean with (12) – SD? Please specify. (SD=12) (equally on page 13 – “82.3% (16)” –> (SD=16) ).

- p. 9: please consider to restructure the results section because now there is a frequent shift from Table 2 to Table 3 and vice versa.

- Table 3: please provide for the variables generation, children and adults gender the information which is the reference category.

- Table 3: please report n for the different immigrant collectives.

- Table 3: the last column (Caribbean) is not visible -> please consider to change the format of this Table to horizontal format.

- Table 4: in the column Attendance please move mean to the line below.

- Table 4: please add the unit (months) to the column Treatment duration.

- S Table: please report the mean age (SD and range) of the children and adults for the different immigrant collectives.

Discussion:

- p. 14: please provide the statement that living in two cultures can also mean benefits for the children (e.g. bi-cultural identity).

- p. 14: please provide also other reasons for the sex-ration differences among adults concerning the utilization of mental health services (e.g. women show a greater openness towards psychotherapy).

- p. 15: please provide an example for traditional theories.

- p. 17: please add the point to the limitations section that because of small sizes of two sub-samples (Europe and Caribbean) inter-ethnic differences probably could not be detected due to insufficient statistical power.

References:

Please translate French titles.

I hope my comments are helpful to the authors and to the editorial team.

6. PLOS authors have the option to publish the peer review history of their article (what does this mean?). If published, this will include your full peer review and any attached files.

Reviewer #1: No

Reviewer #2: Yes: Charlotte Sonne

Reviewer #3: No

---

## [Author Response · Author response to Decision Letter 0]

22 Mar 2020

Please refer to the attached file 'answer to the reviewers'

---

## [Decision Letter · Decision Letter 1]

14 Jul 2020

PONE-D-19-34239R1

An epidemiological description of 529 families referred for French Transcultural Psychotherapy: A decade of experience

PLOS ONE

Dear Dr. Lachal,

Thank you for submitting your manuscript to PLOS ONE. After careful consideration, we feel that it has merit but does not fully meet PLOS ONE’s publication criteria as it currently stands. Therefore, we invite you to submit a revised version of the manuscript that addresses the points raised during the review process.

We look forward to receiving your revised manuscript.

Kind regards,

Stephan Doering, M.D.

Academic Editor

PLOS ONE

Reviewers' comments:

Reviewer's Responses to Questions

**Comments to the Author**

1. If the authors have adequately addressed your comments raised in a previous round of review and you feel that this manuscript is now acceptable for publication, you may indicate that here to bypass the “Comments to the Author” section, enter your conflict of interest statement in the “Confidential to Editor” section, and submit your "Accept" recommendation.

Reviewer #3: (No Response)

Reviewer #4: (No Response)

2. Is the manuscript technically sound, and do the data support the conclusions?

Reviewer #3: Yes

Reviewer #4: Yes

3. Has the statistical analysis been performed appropriately and rigorously? 

Reviewer #3: Yes

Reviewer #4: Yes

4. Have the authors made all data underlying the findings in their manuscript fully available?

Reviewer #3: No

Reviewer #4: Yes

5. Is the manuscript presented in an intelligible fashion and written in standard English?

Reviewer #3: Yes

Reviewer #4: Yes

6. Review Comments to the Author

Reviewer #3: Thank you very much for the revised version of the manuscript. Almost all my suggested improvements have been adequately implemented. There are three minor points that should be modified:

1. In the modified title "the" should be deleted.

2. Abstract: Please delete the word “descriptive” in the first sentence of the method because it can be misunderstood due to the fact that the presented study is not only a descriptive study because it reports not only descriptive statistics.

3. Please report in the text not only the ORs but also the CIs.

Reviewer #4: The authors provide a retrospective analysis of 529 families referred to French Transcultural Psychotherapy. The manuscript has been extensively revised based on previous reviewer comments and has benefitted from the revision. I only have a few minor additional points:

1) There are several grammar mistakes in the manuscript and in particular in the abstract that should be corrected.

2) Some information is scattered through the methods section and the results section a bit. In particular, the information on the time period of included referrals and the time period of data collection is distrubuted in an unfortunate way. I would suggest to summarize the time span of observation as well as the retrospective data collection in one paragraph once and not distributed over different sections.

3) To me it is not clear why the rejected referral group is included in such a purely descriptive way. Wouldn´t a statistical comparison between rejected and treated referrals be a bit more interesting?

7. PLOS authors have the option to publish the peer review history of their article (what does this mean?). If published, this will include your full peer review and any attached files.

Reviewer #3: No

Reviewer #4: No

---

## [Author Response · Author response to Decision Letter 1]

17 Jul 2020

Response to Reviewers

An epidemiological description of 529 families treated by the French method of Transcultural Psychotherapy: A decade of experience

Reviewer #3

Thank you very much for the revised version of the manuscript. Almost all my suggested improvements have been adequately implemented. There are three minor points that should be modified:

1. In the modified title "the" should be deleted.

2. Abstract: Please delete the word “descriptive” in the first sentence of the method because it can be misunderstood due to the fact that the presented study is not only a descriptive study because it reports not only descriptive statistics.

3. Please report in the text not only the ORs but also the CIs.

We deleted “An” in the title and reports CI in the text. We checked but ‘descriptive’ was already deleted in the final abstract, maybe we made a copy/past mistake during the submission process. 

Reviewer #4

The authors provide a retrospective analysis of 529 families referred to French Transcultural Psychotherapy. The manuscript has been extensively revised based on previous reviewer comments and has benefitted from the revision. I only have a few minor additional points:

1) There are several grammar mistakes in the manuscript and in particular in the abstract that should be corrected.

Sorry for that, the entire text was reread by a native speaker. 

2) Some information is scattered through the methods section and the results section a bit. In particular, the information on the time period of included referrals and the time period of data collection is distrubuted in an unfortunate way. I would suggest to summarize the time span of observation as well as the retrospective data collection in one paragraph once and not distributed over different sections.

We moved the time period of data collection near the time span of observation.

3) To me it is not clear why the rejected referral group is included in such a purely descriptive way. Wouldn´t a statistical comparison between rejected and treated referrals be a bit more interesting?

Patients are referred to PTC from other institution, and we do not have access to medical files. The decision of treatment is made after a meeting-call with the therapist who refer the patient. Reason for refusal is the only data available for this group. We agree that comparison between the two group would be interesting, so we add this information in the limits of the study.

Modified text

No descriptive data were available for the rejected referral group, making comparison with the other groups impossible.

---

## [Editor Report · Decision Letter 2]

20 Jul 2020

Epidemiological description of 529 families referred for French Transcultural Psychotherapy: A decade of experience

PONE-D-19-34239R2

Dear Dr. Lachal,

We’re pleased to inform you that your manuscript has been judged scientifically suitable for publication and will be formally accepted for publication once it meets all outstanding technical requirements.

Kind regards,

Stephan Doering, M.D.

Academic Editor

PLOS ONE

---

## [Editor Report · Acceptance letter]

24 Jul 2020

PONE-D-19-34239R2 

Epidemiological description of 529 families referred for French Transcultural Psychotherapy: A decade of experience 

Dear Dr. LACHAL:

I'm pleased to inform you that your manuscript has been deemed suitable for publication in PLOS ONE. Congratulations! Your manuscript is now with our production department. 

Kind regards, 

on behalf of

Professor Stephan Doering 

Academic Editor

PLOS ONE